# Repurposing High-Throughput Screening Reveals Unconventional Drugs with Antimicrobial and Antibiofilm Potential Against Methicillin-Resistant *Staphylococcus aureus* from a Cystic Fibrosis Patient

**DOI:** 10.3390/antibiotics14040402

**Published:** 2025-04-14

**Authors:** Arianna Pompilio, Veronica Lupetti, Valentina Puca, Giovanni Di Bonaventura

**Affiliations:** 1Department of Medical, Oral and Biomedical Sciences, “G. d’Annunzio” University of Chieti-Pescara, 66100 Chieti, Italy; arianna.pompilio@unich.it (A.P.); veronica.lupetti@studenti.unich.it (V.L.); 2Center for Advanced Studies and Technology, “G. d’Annunzio” University of Chieti-Pescara, 66100 Chieti, Italy; 3Department of Pharmacy, “G. d’Annunzio” University of Chieti-Pescara, 66100 Chieti, Italy; valentina.puca@unich.it

**Keywords:** high-throughput screening, *Staphylococcus aureus*, cystic fibrosis, antibacterial, antibiofilm

## Abstract

**Background/Objectives:** Antibiotic therapy faces challenges from rising acquired and biofilm-related antibiotic resistance rates. High resistance levels to commonly used antibiotics have been observed in methicillin-resistant *Staphylococcus aureus* (MRSA) strains among cystic fibrosis (CF) patients, indicating an urgent need for new antibacterial agents. This study aimed to identify potential novel therapeutics with antibacterial and antibiofilm activities against an MRSA CF strain by screening, for the first time, the Drug Repurposing Compound Library (MedChem Express). **Methods/Results:** Among the 3386 compounds, a high-throughput screening-based spectrophotometric approach identified 2439 (72%), 654 (19.3%), and 426 (12.6%) drugs active against planktonic cells, biofilm formation, and preformed biofilm, respectively, although to different extents. The most active hits were 193 (5.7%), against planktonic cells, causing a 100% growth inhibition; 5 (0.14%), with excellent activity against biofilm formation (i.e., reduction ≥ 90%); and 4, showing high activity (i.e., 60% ≤ biofilm reduction < 90%) against preformed biofilms. The potential hits belonged to several primary research areas, with “cancer” being the most prevalent. After performing a literature review to identify other, already published biological properties that could be relevant to the CF lung environment (i.e., activity against other CF pathogens, and anti-inflammatory and anti-virulence potential), the most interesting hits were the following: 5-(N,N-Hexamethylene)-amiloride (diuretic), Toremifene (anticancer), Zafirlukast (antiasthmatic), Fenretide (anticancer), and Montelukast (antiasthmatic) against planktonic *S. aureus* cells; Hemin against biofilm formation; and Heparin, Clemastine (antihistaminic), and Bromfenac (nonsteroidal anti-inflammatory) against established biofilms. **Conclusions:** These findings warrant further in vitro and in vivo studies to confirm the potential of repurposing these compounds for managing lung infections caused by *S. aureus* in CF patients.

## 1. Introduction

*Staphylococcus aureus* is the most prevalent bacterium isolated from the sputum of cystic fibrosis (CF) patients during their first decade of life. The incidence of methicillin-sensitive *S. aureus* (MSSA) is significantly lower than that of methicillin-resistant *S. aureus* (MRSA) in most CF patients [1]. Chronic lung infections caused by methicillin-resistant *S. aureus* (MRSA) strains lead to worse clinical outcomes, including a more rapid decline in lung function (measured as forced expiratory volume in 1 s, FEV_1_) and increased mortality rates [2,3].

Antibiotic therapy presents challenges due to rising rates of antibiotic resistance, particularly among biofilm-related infections [1]. MRSA strains show high resistance to commonly used antibiotics in CF patients, such as ciprofloxacin, clindamycin, erythromycin, and gentamicin. This underscores the pressing need for new antibacterial agents [4].

Traditional drug discovery processes are often lengthy, labor-intensive, and expensive. In contrast, drug repurposing offers a more dynamic, cost-effective, and feasible approach to quickly addressing the declining drug discovery pipeline. This strategy involves exploring new applications for already approved pharmaceuticals, thus reducing the time required for drug development, lowering costs, and minimizing the inherent risks of drug innovation [5].

This study aimed to identify potential novel therapeutics against *S. aureus* by screening the Drug Repurposing Compound Library (MedChem Express) for antibacterial and antibiofilm activities for the first time.

## 2. Results

In this study, we screened the Drug Repurposing Compound Library from Med-Chem Express to identify compounds with antibacterial and antibiofilm activity against *S. aureus*. The library contains 3386 bioactive compounds, including 2342 that have already been launched and 1044 that have reached clinical trial stages in the USA. Among these are 1 drug in Phase I, 606 in Phase II, 372 in Phase III, and 65 in Phase IV (Figure 1A). These compounds have various therapeutic indications in several research areas: 973 for “cancer” (Phase I: 0.1%, Phase II: 30.6%, Phase III: 17.6%, Phase IV: 1.3%, and Launched: 50.4%), 570 for “infection” (Phase II: 7.9%, Phase III: 5.1%, Phase IV: 0.9%, and Launched: 86.1%), 558 for “neurological disease” (Phase II: 15.6%, Phase III: 10%, Phase IV: 1.3%, and Launched: 73.1%), 427 for “inflammation/immunology” (Phase II: 19.2%, Phase III: 8%, Phase IV: 1.6%, and Launched: 71.2%), 109 for “endocrinology” (Phase II: 4.6%, Phase III: 4.6%, Phase IV: 0.9%, and Launched: 89.9%), 298 for “cardiovascular disease” (Phase II: 10.4%, Phase III: 7.7%, Phase IV: 1%, and Launched: 80.9%), and 286 for “metabolic disease” (Phase II: 16.8%, Phase III: 14.3%, Phase IV: 5.6%, and Launched: 63.3%) (Figure 1B).

### 2.1. HTS Assay Validation

The screening quality was assessed using the Z-factor, a standard measure of the robustness and feasibility of a high-throughput screening (HTS). The Z-factor quantifies the difference in response between the positive and negative controls in relation to the combined standard deviations of both controls.

The average Z-factor between the negative and positive controls in the 96-well test plates was 0.675, ranging from 0.500 to 0.843. This value is well above the 0.500 threshold, indicating that the assay can reliably distinguish between positive and negative controls (Figure 2). In addition to the Z-factor, the coefficient of variation was less than 10%, and the signal-to-background ratio exceeded 10-fold, further demonstrating the effectiveness of our anti-*S. aureus* drug screening assay.

### 2.2. Identification of Hits Inhibiting S. aureus Growth

We identified compounds that are effective against the MRSA Sa2 strain through HTS using a 96-well microtiter plate format. A total of 3386 compounds from our compound library were initially tested at a single concentration of 0.1 mM to identify active hits. Based on spectrophotometric measurements taken from the supernatant, the results were expressed as the percentage of bacterial growth inhibition compared to untreated control samples.

This primary screen initially identified 2439 out of 3386 (72%) compounds able to affect *S. aureus* Sa2 growth. Among those, 248 hits had “bacterial” as their primary target and were excluded since this work aimed to find new repositionable drugs. Most (146 out of 248; 58.9%) of these excluded compounds showed high (i.e., 60% ≤ growth reduction < 90%) or excellent activity (i.e., growth reduction ≥ 90%), therefore confirming the robustness of HTS.

The remaining 2191 (64.7%) hits showed potential for repurposing against *S*. *aureus*, although to different extents: 617 (18.2%) with low activity (i.e., 10% < growth reduction < 25%); 809 (23.9%) with moderate activity (i.e., 25% ≤ growth reduction < 60%); 350 (10.3%) with high activity (i.e., 60% ≤ growth reduction < 90%); and, more interestingly, 415 (12.2%) compounds with excellent activity (i.e., growth reduction ≥ 90%).

The 415 hits showing excellent activity based on spectrophotometric readings were re-assessed by cell viable count, confirming the efficacy of 364 (10.7%) compounds, listed in Appendix A and graphed in Figure 3. Among those, it is worth noting that 193 hits caused a 100% inhibition rate. Based on their known pharmacological profiles, the hits were grouped into classes based on the primary research area (some compounds belonged to multiple areas). Most of the hits belonged to “cancer” (83 out of 193; 43.0%) as a primary research area, followed by “infection” (other than bacterial) (31; 16.1%), “neurological disease” (28; 14.5%), “inflammation” (17; 8.8%), “metabolic disease” and “cardiovascular disease” (12 each; 6.2%), “endocrinology”, and “others” (5 each; 2.6%) (Figure 3). Dunnett’s multiple comparison tests found no statistically significant differences in antibacterial activity among research areas.

### 2.3. Identification of Hits Active Against S. aureus Biofilm Formation

The compound library also underwent a 96-well microtiter plate-based HTS to identify hits that could affect biofilm formation by the MRSA Sa2 strain. The results were expressed as the percentage of inhibition of biofilm biomass formation compared to that in untreated controls, estimated from spectrophotometric readings after a crystal violet assay. The compounds significantly affecting biofilm formation, with no known antibacterial activity and inactive against *S. aureus* Sa2 (i.e., growth inhibition ≤ 10%), were considered potential antibiofilm hits.

HTS revealed that 654 out of 3386 (19.3%) compounds can reduce biofilm formation by *S. aureus* Sa2, although to different extents: 128 out of 654 (19.6%) with low activity (i.e., 10% < biofilm reduction < 25%), 344 (52.6%) with moderate activity (i.e., 25% ≤ biofilm reduction < 60%); 168 (25.7%) with high activity (i.e., 60% ≤ biofilm reduction < 90%); and 14 (2.1%) compounds with excellent activity (i.e., biofilm reduction ≥ 90%). Overall, hits belonged to several primary research areas, with “cancer” being the most prevalent, as shown in Figure 4. Dunnett’s multiple comparison tests found no statistically significant differences in activity against biofilm formation among research areas.

The fourteen hits showing excellent activity were retested in a secondary screen—carried out in two independent experiments, each in triplicate—confirming an antibiofilm potential for 5 compounds (Appendix A). These repositionable hit candidates comprised three anticancer agents (Tipifarnib, Olaparib, and Acefylline), Hemin (“cardiovascular disease” research area), and TMC647055 (choline salt) (“infection” research area).

### 2.4. Identification of Hits Active Against Preformed Biofilm by S. aureus

Each of the 3386 compounds was tested at a single concentration of 0.1 mM to pinpoint hits effective towards preformed, 24 h old biofilm formed by the MRSA Sa2 strain. The results were expressed as the percentage of biofilm biomass dispersion compared to that in untreated controls, estimated from spectrophotometric readings after a crystal violet assay. The compounds with significant activity against mature biofilms, with no known antibacterial activity, and not active against *S. aureus* Sa2, were considered potential antibiofilm hits.

The primary screen identified 426 out of 3386 (12.6%) compounds with the potential for repurposing against mature biofilms by *S. aureus* Sa2, although to different extents. Most of the hits showed low (i.e., 10% < biofilm reduction < 25%) (216 out of 426; 56.1%) or moderate (i.e., 25% ≤ biofilm reduction < 60%) (202, 40.7%) activity, whereas only 8 (3.1%) showed high activity (i.e., 60% ≤ biofilm reduction < 90%). No compound was able to disperse at least 90% of biofilm biomass.

Overall, hits belonged to several primary research areas, although those with low and moderate activity mainly belonged to “neurological disease” and “cancer” (Figure 5). Dunnett’s multiple comparison tests found no statistically significant differences in activity against mature biofilm among research areas.

The eight hits showing high activity in the primary screening were re-assessed in a secondary screen—carried out in two independent experiments, each in triplicate—confirming four repositionable hit candidates, comprising Flumatinib (mesylate) with anticancer activity, the anti-inflammatory Bromfenac (sodium hydrate), Clemastine (fumarate) (“neurological disease”), and Heparin (lithium salt) (Appendix A).

## 3. Discussion

In the present study, we screened the Drug Repurposing Compound Library (MedChem Express) to identify hits with relevant potential for antibacterial and antibiofilm activities against an *S. aureus* strain causing a long-term pulmonary infection in a CF patient.

The library we tested in the present study consists of 3386 bioactive compounds, most of which have already been launched and used as drugs with several therapeutic indications, including cancer and neurodegenerative, infectious, and cardiovascular diseases. The library structure could explain the surprisingly high hit rates we observed during the first HTS: 72% (2440 out of 3386) for antibacterial activity, 19.3% (654 out of 3386) for activity against biofilm formation, and 12.6% (426 out of 3386) for activity against preformed biofilm. Indeed, less than 0.1% of hit rates were reported in other HTS studies evaluating large, random chemical libraries of small synthetic molecules that commonly contain numerous non-drug-like molecules [6,7]. The calculation of the Z-factor, the coefficient of variation, and the signal-to-background ratio validated the robustness and feasibility of our anti-*S. aureus* drug HTS assay.

An HTS was initially performed to identify compounds active against planktonic cells of the *S. aureus Sa2* strain. Among the 3386 compounds screened, 2191 compounds—with a primary target different from “bacterial”—caused a decrease in Sa2 strain growth, although with varying effectiveness. We focused on selecting 193 compounds with the highest potential for repurposing since they exhibited maximum activity (i.e., 100% growth reduction), as confirmed by viable cell counts.

A revision of the scientific literature indicated that 135 out of 193 compounds were never reported to be active against *S. aureus* in previous studies and could be further investigated for repurposing potential. In our attempt to individuate drugs with a high repurposing potential relevant to the CF lung infection, the selected 135 potential hits underwent a literature review to reveal other biological effects critical to the interplay between the host and pathogens that shape the course of the disease, i.e., activity against other CF pathogens (e.g., *Pseudomonas aeruginosa*, *Burkholderia cepacia*, *Haemophilus influenzae*) and anti-inflammatory and anti-virulence potential [8]. In this way, 37 compounds were identified and are listed in Table 1.

Here, we reported antibacterial activity for 31 out of 37 selected compounds for the first time. Previous studies referred to the activity of the remaining 6 compounds against species other than *S. aureus*: the anticancer compound Toremifene and the antiasthmatic Zafirlukast vs. *Porphyromonas gingivalis* [9,10]; Zafirlukast, the anticancer Napabucasin, and the antipsychotic Perphenazine vs. oral streptococci [10,11,12]; Zafirlukast and the anticancer compound Linsitinib vs. *Mycobacterium tuberculosis* [13,14]; and Perphenazine vs. *Neisseria meningitidis*, *Enterobacteriaceae*, and *Listeria monocytogenes* [11].

Opportunistic polymicrobial bacterial airway infection is a hallmark of CF lung disease and early mortality [15]. From birth, the airways of individuals with CF are susceptible to infections by microbial opportunists. Over time, more concerning bacterial species often appear in respiratory secretions throughout their shortened lifetimes. The failure of bacterial clearance leads to a dominant chronic inflammation, resulting in a toxic pro-inflammatory local microenvironment that damages the lung and the innate immunity, further facilitating infections and resulting in the predominant cause of morbidity and mortality in CF patients. In this sense, it is worth noting the previously reported activity of the diuretic 5-(N,N-Hexamethylene)-amiloride and Perphenazine, respectively, against *Pseudomonas cepacia* [16] and *H. influenzae* [11], as these species pose the potential to cause pulmonary exacerbations and lung function decline in individuals suffering from CF [17,18].

Other recent evidence suggests that viridans group streptococci (VGS) may play a crucial role in maintaining a stable microbial ecology within the CF lung, enhancing virulence associated with polymicrobial interactions and directly contributing to the pathology of the lung. *Streptococcus pneumoniae* has recently been associated with an increased severe decline in FEV_1_ [19]. In addition, VGS may have the ability to aggregate with other important CF bacterial pathogens, such as *S. aureus* and *P. aeruginosa*, promoting initial attachment and eventual colonization with these pathogens and the development of multispecies biofilms in the CF airways [20]. In this regard, the previously reported activity of Perphenazine against *S. pneumoniae* [11] and Toremifene against *Streptococcus mutans* [9] adds value to their potential for repurposing, warranting further studies.

The prolonged use of antibiotics has been essential in improving the survival rates of CF patients. However, this approach leads to complications, such as increased adaptive antibiotic resistance, adverse effects, and allergic reactions. An attractive alternative recently explored for treating bacterial infections is targeting or reducing the production of virulence factors, such as inhibiting quorum sensing, biofilm formation, iron uptake, and efflux pumps [21,22]. Unlike antimicrobial therapies, this strategy does not affect bacterial growth and is less likely to lead to resistance, exerting low selective pressure. In this sense, five drugs among the hits we identified in the present study have previously been shown to exhibit anti-virulence potential against *S. aureus* and *P. aeruginosa*, increasing their repurposing potential in CF patients (Table 1). *S. aureus* hemolysis activity has been observed to be significantly affected in over 200 clinical isolates by the antihistamine compound Loratadine [23], and by the antiparasitic Miltefosine in the invertebrate *Galleria mellonella* and a murine model of pneumonia [24]. The virulence potential of *S. aureus* is also decreased by the synthetic estrogen Diethylstilbestrol—causing a reduction in alpha-toxin, coagulase, deoxyribonuclease, and penicillinase production [25]—and Loratadine, affecting biofilm formation, pigmentation [23], and exotoxin production [26]. In a rat subcutaneous catheter model, biofilm formation was also inhibited by the anticancer compound Toremifene [27]. Miltefosine and the anti-hypercholesterolemic agent Lovastatin showed anti-virulence potential in *P. aeruginosa*, respectively, inhibiting phospholipase C/sphingomyelinase with a protective effect against murine pneumonia [28] and decreasing swarming motility [29]. Other studies revealed the synergistic activity of Toremifene and the anticancer compound Mitotane combined with polymyxin B against multidrug-resistant *P. aeruginosa* [30,31] and the percutaneous enhancer Laurocapram combined with cephalosporins against MRSA [32].

The main hallmark of CF pathophysiology is excessive inflammation and the inability to resolve lung infections, contributing to morbidity and, eventually, mortality. Therefore, anti-inflammatory properties could be desirable to improve the repurposing potential of the hits found in the present study. Most of the 37 selected hit compounds have been reported for their anti-inflammatory potential in the literature, although in clinical settings or models not pertinent to infectious diseases (e.g., inflammatory bowel disease [33], allergic dermatitis [34], pulmonary arterial hypertension [35], osteoarthritis [36,37], asthma [38], Parkinson’s disease [39], cigarette smoke [40], vascular diseases [41], thyroid eye disease [42], and amyotrophic lateral sclerosis [43]) (Table 1). Conversely, a few hits have previously been reported for anti-inflammatory potential in models that could be relevant to CF lung infections, thus increasing their potential for repurposing: 5-(N,N-Hexamethylene)-amiloride in lipopolysaccharide (LPS)-exposed alveolar epithelial cells [44], Zafirlukast in LPS-exposed mice [45], GW 501,516 (Cardarine)—an aromatic ether under investigation for lipid metabolism’s disorder (phase 2)—and the anticancer compound Ricolinostat, which protect against LPS-activated macrophage inflammation [46,47], Bardoxololone—a semisynthetic triterpenoids under investigation for lymphoma—Omaveloxolone—a potent anti-inflammatory for treating Friedreich’s ataxia—and Verteporfin—used as a photosensitizer in photodynamic therapy—that alleviates LPS-triggered acute lung injury in mice [48,49,50], and Vandertanib—approved for unresectable and disseminated diseases—in SARS-CoV2-infected mice [51] (Table 1). Interestingly, the literature has already reported anti-inflammatory properties in CF patients or models for three hits. Specifically, Fenretide increases IL-1β expression in Cftr-knockout mice, improving their ability to combat *P. aeruginosa* lung infection [52]; in addition, it normalizes the fatty acid imbalance by reducing arachidonic acid and increasing docosahexaenoic acid in CF patients [53]. Zafirlukast, a leukotriene receptor antagonist licensed for asthma prevention, significantly improves the NIH clinical score observed in a pilot study involving CF patients [54], probably due to ameliorated lung tissue pathology and reduced inflammatory cell infiltration reported in LPS-induced lung inflammation, both in vitro and in mice [45]. Finally, the quinoline Montelukast, approved for asthma and allergic rhinitis, decreases eosinophil cationic protein and IL-8 serum and sputum levels, as well as the sputum levels of myeloperoxidase, in CF patients [55].

Bacterial persistence in CF lungs is influenced by local host defense impairments and the pathogens’ ability to adapt to a challenging environment due to selective pressures, such as hyperinflammation, oxidative stress, limited nutrients, anaerobiosis, increased acidity, and antibiotic exposure. Biofilm formation is a key adaptive strategy for bacteria, allowing them to persist even when antibiotics are administered. Currently, CF is considered a biofilm-associated disease, and controlling these biofilms is crucial for effective antimicrobial strategies. Several studies reported direct *S. aureus* biofilm visualization in CF patients by scanning electron microscopy and fluorescence electron in situ hybridization using a specific peptide nucleic acid and further viability evaluation by confocal laser scanning microscopy [56]. Biofilm formation was a common trait of both MRSA and MSSA strains in CF patients, and high biofilm-formation ability has been associated with fewer pulmonary exacerbations and, conversely, exacerbations negatively impacted biofilm formation [57].

In this sense, in the second step of the present study, the compound library also underwent a 96-well microtiter plate-based HTS to identify hits that could affect biofilm formation and preformed, mature, biofilm by the MRSA Sa2 strain. Our findings indicated that 654 compounds affect biofilm formation, although to different extents. Notably, five compounds, i.e., the anticancer Tipifarnib and Olaparib, the bronchodilator Acefylline, Hemin, and the NS5b polymerase inhibitor TMC647055, under investigation for treating hepatitis C—exhibited, for the first time, the highest potential for repurposing due to excellent activity against biofilm formation, which resulted in a reduction of ≥90% compared to the untreated control (Table 2). A careful revision of the literature aimed at identifying additional biological properties of interest in managing CF patients revealed that Olaparib, Acefylline, and Hemin were previously reported for anti-inflammatory activity in a LPS-induced acute lung injury model [58,59,60]. Hemin can also affect *S. aureus* virulence by downregulating hemolysin expression [61]. In disagreement with our findings, Tipifarnib and Hemin have previously been shown to inhibit *S. aureus* growth [62,63].

Regarding hits active against preformed biofilms, four compounds exhibited the highest activity, i.e., 60% ≤ biofilm reduction < 90% (Table 3). The antibiofilm activity exhibited by Clemastine (fumarate), a histamine receptor H1 antagonist, is probably due to a decrease in the transcriptional level of the biofilm formation-relevant *fnbB*, *icaA*, and *icaB* genes in *S. aureus* [64]. Conflicting findings come from the literature concerning the antibiofilm potential of Heparin, a highly sulfated glycosaminoglycan with uneven chain length, routinely used in central venous catheters to prevent thrombosis. Our findings disagree with a previous study reporting that Heparin augments biofilm formation in *S. aureus* [65,66], probably due to extracellular DNA-binding proteins on the *S. aureus* surface that mediate the incorporation of Heparin into the biofilm matrix [67]. A similar pro-biofilm effect was observed for *Escherichia coli* [68], *Staphylococcus epidermidis* [65], and in patients with colorectal cancer where Heparin increased the biofilm formed by intestinal flora, thus providing a protective layer in the intestinal tract of patients [69]. In agreement with our findings, heparinoids, glycosaminoglycans chemically and pharmacologically related to Heparin, were found to suppress biofilm formation in *Cutibacterium acnes* by inhibiting the AI-2-mediated QS [70]. No evidence of activity against preformed biofilms has been previously published for the anticancer compound Flumatinib (mesylate) and the nonsteroidal anti-inflammatory drug Bromfenac (sodium hydrate). All five hits showing high potential for dispersing established biofilm have also been reported to trigger anti-inflammatory activity [71,72,73,74], although only for Heparin under experimental settings relevant to CF patients, such as human bronchial cells [72] and LPS-induced lung injury [73].

**Table 1 antibiotics-14-00402-t001:** Compounds (n = 37) showing maximum activity against planktonic cells (i.e., 100% growth reduction) of S. aureus Sa2 at HTS. Classification, therapeutic category, and mechanism of action are provided, along with other biological properties relevant to cystic fibrosis (CF).

Compound	Classification Therapeutic Category and Indication(s) Mechanism(s) of Action	Antibacterial Activity	Anti-Virulence Activity	Anti-Inflammatory Activity
Perphenazine	■ Phenothiazine; ■ Approved for schizophrenia, psychosis, nausea, and vomiting; investigational for agitation (phase 3), cocaine dependence (phase 2), and depressive disorder (phase 3); ■ D(2) and D(1A) dopamine receptor antagonist.	■ *Neisseria meningitidis, Haemophilus influenzae, Enterobacteriaceae, Streptococcus pneumoniae*, group B streptococci, *Listeria monocytogenes* [11]		■ Inhibition of infiltrated mast cells into the lesion area [34].
Miltefosine	■ Hexadecyl monoester of phosphocholine; ■ Approved for trypanosomiasis and cutaneous, mucocutaneous, and visceral leishmaniasis; investigational for urticaria (phase 2); ■ Phospholipase A2 inhibitor.		■ Reduction in *S. aureus* hemolysis activity in a murine infection model [24]; ■ Protective effect vs. murine pneumonia by inhibiting *P. aeruginosa* phospholipase C/sphingomyelinase activity [28].	■ T-cell proliferation inhibition [33].
Diethylstilbestrol	■ Synthetic nonsteroidal estrogen; ■ Approved for menopausal and postmenopausal disorders, neoplasm; investigational for prostate cancer (phase 3); ■ Estrogen receptor beta agonist.		■ Reduction in *S. aureus* alpha-toxin, coagulase, deoxyribonuclease, and penicillinase [25].	■ Inhibition of phospholipase D activity and degranulation by stimulated human neutrophils [75].
Selexipag	■ Pyrazine; ■ Approved for group 1 pulmonary arterial hypertension; ■ Prostacycline receptor agonist.			■ Anti-inflammatory potential [35].
AZD-9056	■ Phenylpropylamine; ■ Investigational for rheumatoid arthritis (phase 2); ■ P2X purinoceptor 7 antagonist.			■ In vivo reduction in expression of IL-1β, IL-6, TNF-α, MMP-13, SP, and PGE2 [37]; ■ Reduction in lipoxin A4 (LXA4), resolvin D1 (RvD1), and 15(S)-hydroxyeicosatetraenoic acid (15(S)-HETE) by alveolar macrophages [38].
Vortioxetine	■ N-arylpiperazine; ■ Approved for major depressive disorders; ■ 5-hydroxytryptamine receptor 3A and 7 antagonist, Sodium-dependent serotonin transporter inhibitor.			■ Anti-oxidative and immunomodulatory effects by directing macrophages towards the alternative phenotype [76].
Lovastatin	■ Fatty acid ester; ■ Approved for hyperlipidemia; ■ 3-hydroxy-3-methylglutaryl-coenzyme A reductase inhibitor.		■ Decrease in *P. aeruginosa* swarming motility [29].	■ Reduction in acute mucosal inflammation via 15-epi-lipoxin A4 [77]; ■ Macrophage and lymphocyte recruitment reduction in a murine model of the whole lung irradiated [78]; ■ Decreased pro-inflammatory cytokine levels in lovastatin-treated respiratory syncytial virus-infected cells [79].
Fenretinide	■ Synthetic retinoid; ■ Investigational for several types of cancer (phases 2–3), lymphoma, and leukemia (phase 1); ■ Induction of cancer cell apoptosis.			■ Decreased IL-1β and S100A8 expression, improving Cftr-knockout mice’s ability to combat *P. aeruginosa* lung infection [52]; ■ Arachidonic acid downregulation and increased levels of docosahexaenoic acid in CF patients [53]; ■ Reduced IL-1β, IL-6, and PGE2 pro-inflammatory cytokine expression via suppression of JAK-STAT, PI3K-Akt, PKC, and downstream NF-κB signaling pathways in *A. actinomycetemcomitans*-infected murine monocyte/macrophage [80].
Napabucasin	■ Naphthofuran; ■ Investigational for colorectal and pancreatic carcinoma (phase 3), glioblastoma multiforme (phase 1), and hematopoietic and lymphoid cell neoplasm (phase 1); ■ Signal transducer and activator of transcription 3 inhibitor.	■ Oral streptococci [12].		■ Reduced TNF-α and IL-6 levels in neuronal rat cells with post-isolation damage [81].
Loratadine	■ Benzocycloheptapyridine; ■ Approved for symptoms of allergic rhinitis, wheal formation, urticaria, and other allergic dermatologic conditions; ■ Histamine H1 receptor antagonist.		■ Reduced *S. aureus* biofilm formation, pigmentation, hemolysis [23], and exotoxin production [26].	■ Inhibition of the release of GM-CSF and IL-8 in A549 human airway epithelial cells [82].
Toremifene	■ Tertiary amine; ■ Approved for metastatic breast cancer; investigational as a preventative agent for prostate cancer (phase 2); ■ Estrogen receptor modulator.	■ *Porphyromonas gingivalis, Streptococcus mutans* [9].	■ *S. aureus* biofilm formation inhibition in a rat subcutaneous catheter model [27]; ■ Synergy with polymyxin B against resistant *P. aeruginosa* [30].	
Etifoxine	■ Benzoxazine; ■ Approved as an anxiolytic and anticonvulsant; ■ Modulation of GABAergic neurotransmission and neurosteroid synthesis.			■ Reduction in inflammatory mediators and infiltration of leukocytes in the brain [39].
5-(N,N-Hexamethylene)-amiloride	■ Pyrazine; ■ Approved as diuretic; ■ Na+/H+ exchangers blocker, apoptosis inducer, antineoplastic agent.	■ *Pseudomonas cepacia* [16].		■ Downregulation of inflammatory signals and upregulation of anti-inflammatory response by targeting the kappaB-α/NF-kappaB signaling transduction pathway in the alveolar epithelium [44].
Zafirlukast	■ Indole; ■ Approved for asthma; investigational for breast and ovarian cancer (phase 2); ■ Leukotriene D4 receptor antagonist.	■ *P. gingivalis, S. mutans* [10]; ■ *Mycobacterium tuberculosis* [13].		■ In vitro and in vivo inflammatory response suppression of alveolar epithelial cells via reduction in the TLR4/NF-κB/NLRP3 inflammasome pathway [45]; ■ Improved clinical score of adult CF patients [54].
Mitoquinone (mesylate)	■ Organophosphorus compound; ■ Investigational (phase 2) for Parkinson’s disease, chronic hepatitis C virus infection, dilated cardiomyopathy, non-alcoholic fatty liver disease, and ulcerative colitis; ■ TPP-based, mitochondrially targeted antioxidant to protect against oxidative damage.			■ Attenuated inflammation, mucus hypersecretion, and oxidative stress induced by cigarette smoke by modulating mitochondrial function and the NF-κB signal pathway [40].
Crisaborole	■ Benzoxaborole; ■ Approved for onychomycosis and mild-to-moderate atopic dermatitis; ■ 3′,5′-cyclic-AMP phosphodiesterase 4 inhibitor.			■ Inhibition of PDE4 leads to elevated levels of cAMP that inhibit the NF-kB pathway and suppress the release of pro-inflammatory mediators [41].
Linsitinib	■ Quinoline; ■ Investigational for Ewing sarcoma (phase 2), colorectal carcinoma (phase 1), head and neck malignant neoplasia (phase 2), hepatocellular carcinoma (phase 2), and ovarian cancer (phase 1); ■ Insulin-like growth factor 1 receptor (IGF-1R) and insulin receptor inhibitor.	■ *M. tuberculosis* [14].		■ Inhibition of the development and progression of thyroid eye disease via infiltration of T lymphocytes and macrophages suppression in a murine model of Graves’ disease [42].
GW 501516 (Cardarine)	■ Aromatic ether; ■ Investigational for the disorder of lipid metabolism (phase 2); ■ Peroxisome proliferator-activated receptor delta (PPARδ) agonist.			■ Suppressed inflammatory mediators with protection vs. LPS-induced macrophage inflammation and acute liver failure in mice [47].
Darapladib	■ Substituted pyrimidone; ■ Investigational for acute coronary syndrome (phase 3), atherosclerosis (phase 3), and diabetic retinopathy (phase 2); ■ Lipoprotein-associated phospholipase-A2 (Lp-PLA2) inhibitor.			■ Reduced expression of nucleotide-binding oligomerization domain-like receptor with pyrin domain 3 (NLRP3) and IL-1β; improved macrophage migration and IL-1β secretion in macrophages by blocking NLRP3 inflammasome activation [83].
Bardoxolone	■ Synthetic triterpenoid; ■ Investigational for lymphoma (phase 1); ■ Nitric oxide synthase inhibitor.			■ NF-κB signaling suppression in mice with LPS-induced acute lung injury [48]; ■ NLRP3 inflammasome and pro-caspase-1 complex assembly inhibition in mice with acute lung injury [84].
Meisoindigo	■ Second-generation derivative of indirubin; ■ Approved for chronic myelogenous leukemia; investigational for active ulcerative colitis (phase 2); ■ Inhibition of DNA biosynthesis and the assembly of microtubules in cancer cells.			■ Inhibition of zebrafish leukocyte chemotactic migration [85]; ■ Blocked activation of the NLRP3 inflammasome and modulation of the polarization of microglia/macrophages via inhibition of the TLR4/NF-κB signaling pathway [86].
Epinastine	■ Benzazepine; ■ Approved for preventing itching associated with allergic conjunctivitis; investigational for allergic rhinitis (phase 3) and seasonal allergic rhinitis (phase 3). ■ Histamine H1 receptor antagonist.			■ Suppression of IL-6 expression in LPS-treated macrophages [87].
Salirasib	■ Salicylic acid derivative; ■ Investigational for non-small-cell lung carcinoma (phase 2); ■ Serine/threonine-protein kinase mTOR modulator; GTPase NRas antagonist.			■ Inhibition of CD4+ and CD8+ T cell proliferation and IFN-γ, TNF-α, IL-6, and IL-17 release in rheumatoid arthritis [36]; ■ Reduced iNOS expression induced by LPS in murine macrophages [88].
Omaveloxolone	■ Semi-synthetic triterpenoid; ■ Approved for Friedreich’s ataxia; investigational for breast cancer (phase 2), liver disease (phase 1), melanoma (phase 1); ■ Nuclear factor erythroid 2-related factor 2 activator.			■ Improvement of LPS-induced acute murine lung injury by interfering with Bach1-mediated ferroptosis [49].
Incyclinide	■ Chemically modified tetracycline; ■ Investigational for central nervous system neoplasm (phase 1); ■ 72 kDa type IV collagenase inhibitor.			■ IL-12 and TNF-α inhibition in chronic periodontitis [89].
Glecaprevir	■ Aminoisobutyric acid; ■ Approved for chronic hepatitis C virus (HCV); ■ NS3 protease HCV inhibitor, genome polyprotein HCV inhibitor.			■ Anti-inflammatory potential [90].
Isradipine	■ Dihydropyridine; ■ Approved for hypertension and cardiovascular disease; investigational for Parkinson’s disease (phase 3), bipolar disorder (phase 2), drug dependence (phase 1), and nicotine dependence (phase 1); ■ T and L type calcium channel inhibitor.			■ PMN leukocyte function inhibition by interference with the adenosine system [91].
Laurocapram	■ Caprolactam; ■ Approved as a percutaneous enhancer; ■ Interaction with lipids in the stratum corneum, thus enhancing skin absorption of a hydrophilic chemical.		■ Improved cephalosporin’s activity against methicillin-resistant *S. aureus* [32].	
Masitinib	■ Benzamide; ■ Approved for amyotrophic lateral sclerosis, mastocytosis, and advanced pancreatic cancer; ■ Proto-oncogene tyrosine-protein kinase Src inhibitor.			■ Reduced survival, migration, and degranulation of mast cells [43].
Mitotane	■ Isomer of the insecticide dichlorodiphenyldichloroethane; ■ Approved for adrenal cortex carcinoma and neoplasm; Investigational for carcinoma (phase 3) and prostate cancer (phase 1); ■ Cytochrome P450 11B1 mitochondrial inducer, estrogen receptor binder.		■ Synergy with polymyxin B against resistant *Acinetobacter baumannii*, *P. aeruginosa*, and *Klebsiella pneumoniae* [31].	
Montelukast	■ Quinoline; ■ Approved for asthma and allergic rhinitis; investigational for bronchiolitis (phase 2), sepsis (phase 2), leukemia (phase 2), sleep apnea (phase 2), non-small cell lung carcinoma (phase 2), premature birth (phase 2), HIV (phase 1), pain (phase 3); ■ Cysteinyl leukotriene receptor 1 antagonist.			■ Decreased eosinophil cationic protein and IL-8 serum and sputum levels, and sputum levels of myeloperoxidase in CF patients [55]; ■ Decreased IL-6, IL-1β, and IL-17 levels in the lung of septic mice [92].
Ricolinostat	■ Pyrimidinecarboxylic acid; ■ Investigational for breast carcinoma (phase 1), chronic lymphocytic leukemia (phase 1), diabetic neuropathy (phase 2), multiple myeloma (phase 1), peripheral nervous system disease (phase 1), hilar cholangiocarcinoma (phase 1), and lymphoma (phase 1); ■ Histone deacetylase 6 (HDAC6) inhibitor.			■ Inhibition of ROS overproduction and TNF-α, IL-1β, and IL-6 expression in LPS-activated RAW264.7 cells [46].
Vandetanib	■ Quinazoline; ■ Approved as an alternative to local therapies for both unresectable and disseminated disease; ■ Vascular endothelial growth factor receptor 2 inhibitor, Proto-oncogene tyrosine-protein kinase receptor Ret inhibitor, epidermal growth factor receptor inhibitor.			■ Reduced IL-6, IL-10, and TNF-α levels and inflammatory cell infiltrates in the lungs of SARS-CoV-2-infected animals [51].
Verteporfin	■ Benzoporphyrin derivative; ■ Approved as a photosensitizer in photodynamic therapy; ■ Photoabsorption.			■ In vitro and in vivo excellent biocompatibility, safety, and anti-inflammatory responses by suppressing M1 macrophage polarization while inducing M2 polarization in acute lung injury [50]; ■ Enhanced PMN-mediated *S. aureus* killing [93].
Vigabatrin	■ Gamma-aminobutyric acid analog; ■ Approved for refractory seizures and infantile spasms; ■ 4-aminobutyrate aminotransferase mitochondrial inhibitor.			■ PMN migration inhibition [94].
Diroximel (fumarate)	■ Fumarate; ■ Approved for immune system disease, multiple sclerosis, and relapsing-remitting multiple sclerosis; ■ Neuronal acetylcholine receptor subunit alpha-10 agonist.			■ Reduced iNOS, COX-2, TNF-α, IL-6, and IL-1β expression in the BV-2 microglial cell line [95].
Infigratinib	■ Phenylurea compound; ■ Approved for advanced or metastatic cholangiocarcinoma, urothelial carcinoma, and achondroplasia; ■ Fibroblast growth factor receptors 1-2-3 inhibitor.			■ Reduced infiltration of lymphocytes and macrophages in a murine model of multiple sclerosis [96].

Data source: PubChem (https://pubchem.ncbi.nlm.nih.gov/, accessed on 9 April 2025). The literature review was performed using PubMed (https://pubmed.ncbi.nlm.nih.gov/, accessed on 9 April 2025) using the following keywords: [compound name] and “antibacterial”, “aureus”, “aeruginosa”, “inflammatory”, “inflammatory lung”, “virulence”, and “cystic fibrosis”.

**Table 2 antibiotics-14-00402-t002:** Compounds (n = 5) showing excellent activity against biofilm formation (i.e., biofilm reduction ≥ 90%) by *S. aureus* Sa2 at HTS. Classification, therapeutic category, and mechanism of action are provided, along with other biological properties relevant to cystic fibrosis (CF).

Compound	Classification Therapeutic Category and Indication(s) Mechanism(s) of Action	Antibacterial Activity	Anti-Virulence Activity	Anti-Inflammatory Activity
Tipifarnib	■ Nonpeptidomimetic quinolone; ■ Investigational for colorectal cancer, leukemia (myeloid), pancreatic cancer, and solid tumors (phases 1, 2, 3); ■ Farnesyltransferase inhibitor.	■ *Staphylococcus aureus, Staphylococcus epidermidis, Streptococcus pneumoniae* [63].		■ Reduced LPS-induced IL1β, IL18, and TNFα secretion in a cellular model of mevalonate kinase deficiency [97].
Olaparib	■ N-acylpiperazine; ■ Approved for treating ovarian cancer, breast cancer, pancreatic cancer, and prostate cancer; ■ Poly (ADP-ribose) polymerase (PARP) inhibitor.			■ Ameliorated LPS-induced acute lung injury by downregulation of TNF-α, IL-1β, and VCAM-1 expression [58].
Acefylline	■ Theophylline derivative; ■ Approved for treating asthma, emphysema, acute and chronic bronchitis associated with bronchospasm; investigational for airway obstruction (phase 4); ■ Adenosine receptor antagonist.	■ *Mycobacterium tuberculosis* [98].		■ Reduced NO, TNF-α, IL-1β, and IL-6 levels in LPS-induced RAW 264.7 cells [59].
Hemin	■ Iron-containing porphyrin; ■ Approved for acute intermittent porphyria; investigational for diabetes mellitus, myocardial ischemia, graft failure (phase 2); ■ 5-aminolevulinate synthase, non-specific, mitochondrial inhibitor.	■ *S. aureus* [62].	■ Downregulation of β- and γ-hemolysins expression in *S. aureus* [61].	■ Attenuated LPS-induced lung injury in mice by inhibiting the activation of the nucleotide-binding domain-like receptor protein 3 (NLRP3) inflammasome and oxidative stress [60].
TMC647055 (Choline salt)	■ Macrolactam; ■ Investigational for treating hepatitis C (phase 1), and chronic hepatitis C (phase 2); ■ Nonnucleoside Hepatitis C Virus NS5B polymerase inhibitor.			

Data source: PubChem (https://pubchem.ncbi.nlm.nih.gov/, accessed on 9 April 2025). The literature review was performed using PubMed (https://pubmed.ncbi.nlm.nih.gov/, accessed on 9 April 2025) using the following keywords: [compound name] and “antibacterial”, “aureus”, “aeruginosa”, “inflammatory”, “inflammatory lung”, “virulence” and “cystic fibrosis”.

**Table 3 antibiotics-14-00402-t003:** Compounds (n = 4) showing activity against established biofilm (i.e., 60% ≤ biofilm dispersion < 90%) by *S. aureus* Sa2 at HTS. Classification, therapeutic category, and mechanism of action are provided, along with other biological properties relevant to cystic fibrosis (CF).

Compound	Classification Therapeutic Category and Indication(s) Mechanism(s) of Action	Antibacterial Activity	Anti-Virulence Activity	Anti-Inflammatory Activity
Clemastine (fumarate)	■ Ethanolamine-derivative; ■ Approved for symptomatic relief of allergic rhinitis, and mild, uncomplicated allergic skin manifestations of urticaria and angioedema; investigational for chronic progressive and relapsing multiple sclerosis (phase 1), urticaria and optic neuritis (phase 2), dermatitis (phase 3), and allergic diseases (phase 4); ■ Histamine receptor H1 (HRH1) antagonist.		■ Inhibition of biofilm formation and hemolytic activity in *S. aureus* [64].	■ Anti-inflammatory and anti-pyroptotic actions via inhibition of the NLRP3 inflammasome in an autoimmune encephalomyelitis rat model [74].
Heparin	■ Sulfated glycosaminoglycan; ■ Approved for preventing blood clots; ■ Antithrombin-III potentiator; coagulation factor X inhibitor.	■ *Staphylococcus aureus, Pseudomonas aeruginosa, Escherichia coli, Streptococcus pneumoniae, Haemophilus influenzae* [99,100].	■ Improved phenol-soluble modulins fibrillation and biofilm formation in *S. aureus* [66]; ■ Augmented biofilm formation by intestinal microbiota in patients with colorectal cancer [69]; ■ Critical for host tissue colonization and invasion of several bacterial pathogens [101].	■ Decreased COX-2 and CXCL-8 gene expression in human bronchial epithelial cells [72]; ■ Reduced IL-6 and TNF-α pulmonary levels in a rat model of LPS-induced acute lung injury [73].
Flumatinib (mesylate)	■ Pyridinylpyrimidine; ■ Investigational for acute lymphoblastic leukemia (phase 3), and chronic myelogenous leukemia (phase 3); ■ Tyrosine-protein kinase ABL1 inhibitor.			
Bromfenac (sodium hydrate)	■ Nonsteroidal anti-inflammatory drug; ■ Approved for treating postoperative eye inflammation; ■ Prostaglandin G/H synthase 1 and 2 inhibitor.	■ *S. aureus, E. coli, Bacillus subtilis* [102].		■ COX-2 inhibition in rabbit retinochoroidal tissues [71].

Data source: PubChem (https://pubchem.ncbi.nlm.nih.gov/, accessed on 9 April 2025). The literature review was performed using PubMed (https://pubmed.ncbi.nlm.nih.gov/, accessed on 9 April 2025) using the following keywords: [compound name] and “antibacterial”, “aureus”, “aeruginosa”, “inflammatory”, “inflammatory lung”, “virulence” and “cystic fibrosis”.

## 4. Materials and Methods

### 4.1. Compound Library

The “Drug Repurposing Compound Library” was purchased from MedChem Express (cat. no. HY-L035; Monmouth Junction, NJ, USA). The library was provided in a 96-well plate format with aliquots of 10 mM stocks of drugs in DMSO or water, stored at −80 °C.

### 4.2. Bacterial Strain and Growth Conditions

HTS was conducted using the *S. aureus* Sa2, an MRSA strain isolated from a respiratory specimen in a CF patient, which caused chronic infection. Some colonies were grown on Tryptone Soy Agar (TSA; Oxoid, Milan, Italy) after overnight incubation at 37 °C. These colonies were then suspended in sterile saline 0.9% to achieve an optical density at 550 nm (OD_550_) of 0.3. This suspension was diluted 1:10 in sterile saline to reach a final 1–2 × 10^7^ CFU/mL concentration. This standardized inoculum was utilized for all assays unless otherwise indicated.

### 4.3. Antibacterial HTS Assay

The library was screened at a single concentration point to identify antibacterial compounds effective against the *S. aureus* Sa2 strain. Specifically, 5 µL of a standardized inoculum (corresponding to 0.5–1 × 10^5^ CFU/well) was added to each well of a 96-well polystyrene microtiter plate containing 100 µL of Tryptone Soy Broth (TSB; Oxoid) along with 1 µL of a 10 mM compound stock solution from the MedChem library. This setup achieved a final drug concentration of 0.1 mM. Uninoculated samples containing 1% (*v*/*v*) DMSO (the final background concentration in each well) were used as blanks. A negative control was also prepared with 50% (*v*/*v*) DMSO to ensure 100% killing of the bacteria. The contents of each well were mixed by pipetting, and the plates were incubated at 37 °C in an aerobic atmosphere. After a 24 h incubation period, the survival rate of the planktonic cells was assessed spectrophotometrically by measuring the OD_550_ of broth culture supernatants using a microplate reader (Tecan Infinite^®^ M PLEX; Tecan Group Ltd., Mannedorf, Switzerland). This value was corrected by subtracting the average OD_550_ value of the uninoculated wells (blanks).

The growth rate percentage was calculated by comparing it to the inoculated control sample (which had not been treated), set at 100% growth. The antibacterial activity of the compounds in the library was categorized based on the percentage of growth reduction compared to the untreated control sample as follows: (i) low efficacy—10% < x < 25%; (ii) moderate efficacy—25% ≤ x < 60%; (iii) high efficacy—60% ≤ x < 90%; and (iv) excellent efficacy—90% ≤ x ≤ 100%. Only drugs causing a ≥90% reduction in bacterial burden, subsequently confirmed by cell viable count, were considered potential anti-*S. aureus* hit compounds and underwent further characterization. In the cell viable count, serial dilutions of the sample prepared in sterile saline were plated on TSA; after incubation for 24 h at 37 °C, cell viability was determined by CFU counting. DMSO at the final concentration of 0.1 mM did not show any activity against *S. aureus* Sa2, as assessed by cell viable count, thus indicating that the antibacterial effects observed were due to the compound only.

### 4.4. HTS Assay Validation

The results from each HTS microplate were validated by calculating the Z-factor. To validate the degree of separation, the Z-factor and the percent inhibition of the positive and negative controls were determined using the following formula:$$Z-factor=1-\frac{3(\sigma_{p}+\sigma_{n})}{\left|\mu_{p}-\mu_{n}\right|}$$
where *σ_p_* and *σ_n_* are the standard deviations of the positive and negative controls, respectively, and *μ_p_* and *μ_n_* are the corresponding mean values. A Z-factor between 0.5 and 1.0 indicates an excellent assay and statistically reliable separation between the positive and negative controls.

### 4.5. Biofilm Inhibition and Dispersion HTS Assays

To assess the efficacy of the drugs in preventing biofilm formation, a standardized inoculum of 5 µL (1–2 × 10^7^ CFU/mL) was added to each TC-treated microplate (Falcon; Diagramma, Città Sant’Angelo, Italy) well. Each well contained 100 µL of TSB with 1 µL of a 10 mM compound stock solution from the MediChem library, achieving a final drug concentration of 0.1 mM. Control samples were prepared with TSB only. The microplate was incubated at 37 °C for 24 h, under an aerobic atmosphere and statically, after which planktonic cells were gently removed by washing each well twice with 100 µL PBS (Sigma-Aldrich Italia s.r.l.; Milan, Italy). The samples were then fixed at 60 °C for 1 h, and biofilm biomass was quantified spectrophotometrically. For the quantification, 100 µL of 1% (*w*/*v*) Hucker’s crystal violet (Sigma-Aldrich s.r.l.) was added to each well. After a 5 min incubation at room temperature, the wells were washed with tap water. Following drying at 37 °C, 100 µL of 33% (*v*/*v*) glacial acetic acid (Sigma-Aldrich Italia s.r.l.) was added to dissolve the stained dye for 15 min. The biofilm biomass was measured by recording the absorbance at 492 nm using a Tecan Infinite^®^ M PLEX microplate reader.

To evaluate the effectiveness of the drugs against preformed biofilms, 24 h old biofilms were treated with 100 µL of TSB containing the selected drug at a final concentration of 0.1 mM, or with TSB only for the controls. These biofilm samples were incubated at 37 °C for another 24 h, under an aerobic atmosphere and statically, and washed with PBS. Finally, biofilm biomass was measured using the crystal violet staining method as previously described.

A potential “antibiofilm hit” had to affect biofilm formation or disperse preformed biofilms, whereas it had no antibacterial activity (growth reduction ≤ 10%). The magnitude of the hit’s activity against biofilm formation or preformed biofilms was categorized as follows: low (i.e., 10% < biofilm reduction < 25%), moderate (i.e., 25% ≤ biofilm reduction < 60%), high (i.e., 60% ≤ biofilm reduction < 90%), and excellent (i.e., biofilm reduction ≥ 90%).

### 4.6. Statistical Analysis

All assays were conducted in triplicate and repeated twice (n = 6). Statistical analysis was performed using GraphPad Prism software (version 7.0; GraphPad Software, San Diego, CA, USA). The data were tested for normal distribution using the Shapiro–Wilk test. Ordinary one-way ANOVA with Dunnett’s multiple comparisons test was applied when assessing differences among three or more groups of unpaired data. The statistical analysis assumed a confidence level of ≥95%, thus considering *p* values < 0.05 statistically significant.

## 5. Conclusions

The present study aimed to identify compounds with high repurposing potential against planktonic and sessile cells of an *S. aureus* strain causing chronic lung infection in a CF patient. In this regard, the HTS of a library compound performed in 96-well microtiter plates with a spectrophotometric assessment of the results revealed a helpful platform. We identified 193 hits causing 100% killing of planktonic cells, 5 causing biofilm reduction of at least 90%, and 4 effectively dispersing from 60 to 90% of established biofilm. Most hits were anticancer agents, followed by anti-infective (other than antibacterial) and anti-inflammatory drugs. Also considering other, already published, biological properties that could be relevant to the CF lung environment, among the hits active against planktonic *S. aureus* cells, those with the most interesting repurposing potential were 5-(N,N-Hexamethylene)-amiloride (active against other CF pathogens and with anti-inflammatory potential), Toremifene (active against other CF pathogens and with anti-virulence potential), and Zafirlukast, Fenretide, and Montelukast (with anti-inflammatory properties already observed in CF patients). Among hits affecting biofilm formation, Hemin also showed anti-virulence and anti-inflammatory effects. Heparin, Clemastine, and Bromfenac are the most interesting hits that significantly affect established biofilms.

Further in vitro and in vivo studies are needed to confirm the potential of repurposing these compounds for managing *S. aureus* lung infections in CF patients.

## Figures and Tables

**Figure 1 antibiotics-14-00402-f001:**
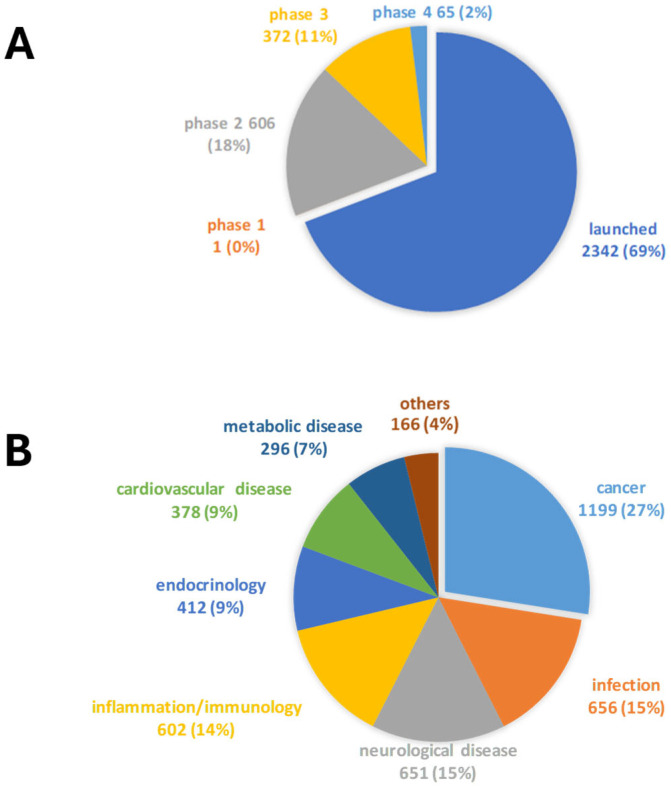
Drug Repurposing Compound Library (MedChem Express) structure: clinical information (**A**) and research area (**B**). Reported are the number of drugs and the relative percentages. Some drugs belong to more research areas.

**Figure 2 antibiotics-14-00402-f002:**
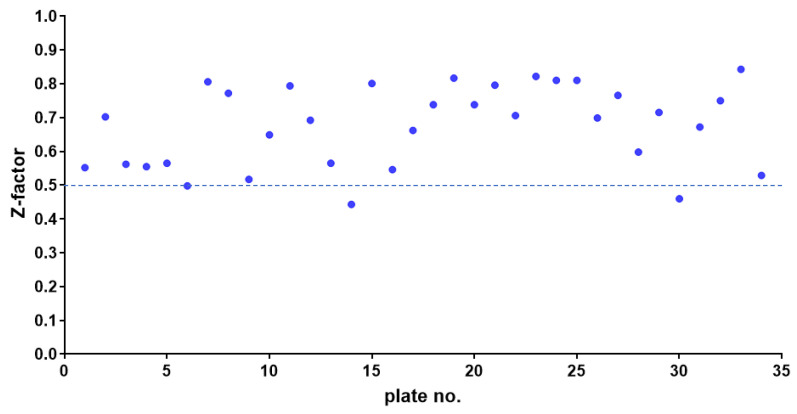
Z-factor plot from high-throughput screening performed in 96-well plates. Solid dots represent the Z-factor from thirty-four 96-well plates. The dashed line indicates a threshold of 0.5.

**Figure 3 antibiotics-14-00402-f003:**
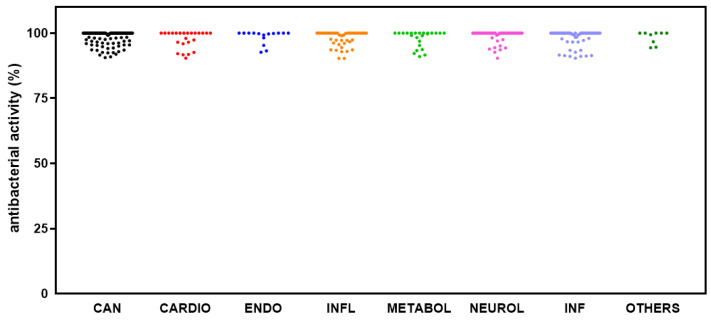
HTS of the compound library to identify active hits against planktonic *S. aureus* Sa2 cells. The antibacterial activity of compounds in several research areas (CAN, cancer; CARDIO, cardiovascular; ENDO, endocrinology; INFL, inflammation; METABOL, metabolic disease; NEUROL, neurological disease; INF, infection; OTHERS, various research areas) is shown as a percentage, referred to as the untreated control.

**Figure 4 antibiotics-14-00402-f004:**
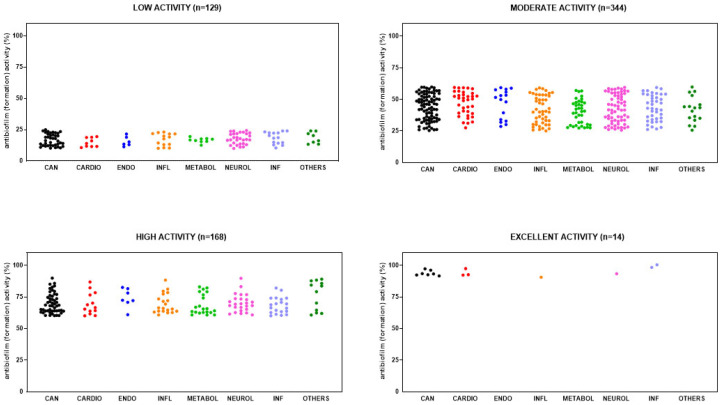
HTS of the compound library to identify active hits against biofilm formation by *S. aureus* Sa2 strain. The antibiofilm activity of compounds in several research areas (CAN, cancer; CARDIO, cardiovascular; ENDO, endocrinology; INFL, inflammation; METABOL, metabolic disease; NEUROL, neurological disease; INF, infection; OTHERS, various research areas) is shown as a percentage referred to as the untreated control. The magnitude of the antibiofilm activity is categorized as follows: low (i.e., 10% < biofilm reduction < 25%), moderate (i.e., 25% ≤ biofilm reduction < 60%), high (i.e., 60% ≤ biofilm reduction < 90%), and excellent (i.e., biofilm reduction ≥ 90%).

**Figure 5 antibiotics-14-00402-f005:**
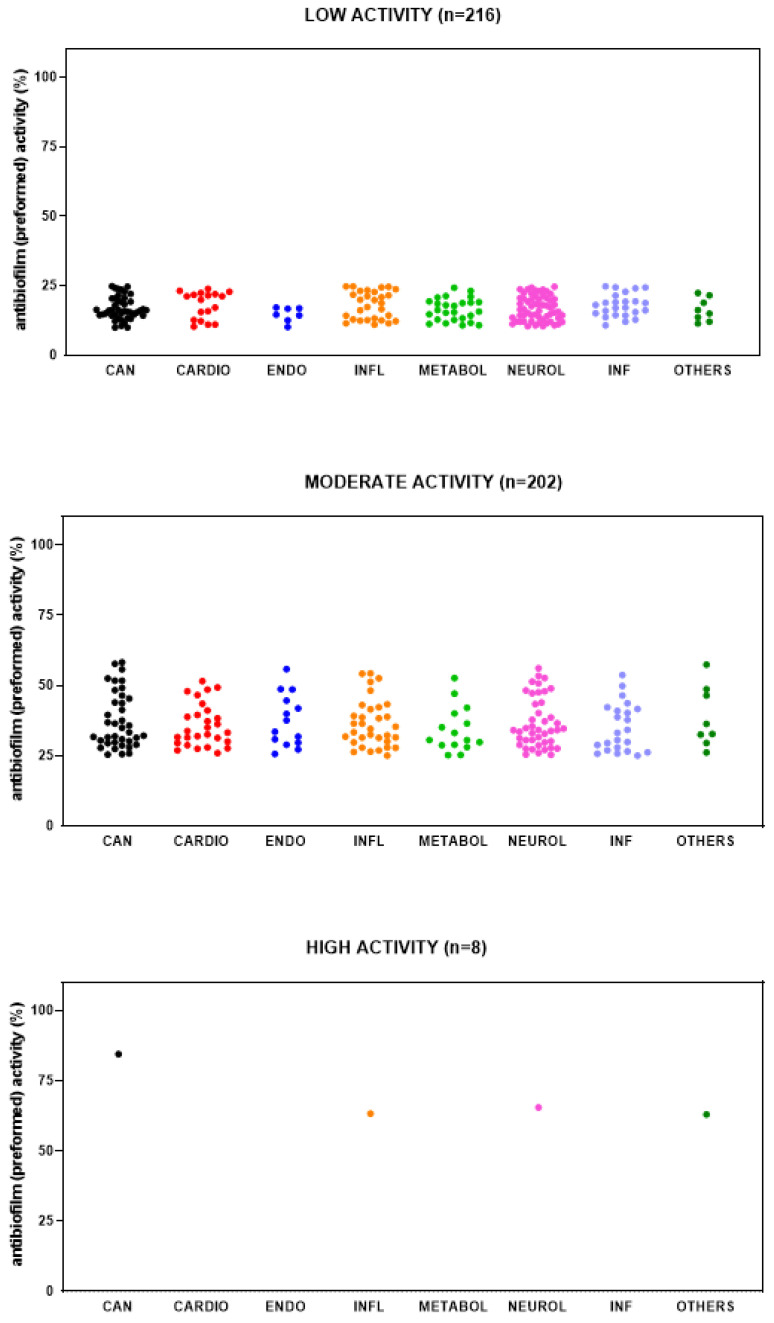
HTS of the compound library to identify active hits against preformed biofilm by *S. aureus* Sa2 strain. The antibiofilm activity of compounds in several research areas (CAN, cancer; CARDIO, cardiovascular; ENDO, endocrinology; INFL, inflammation; METABOL, metabolic disease; NEUROL, neurological disease; INF, infection; OTHERS, various research areas) is shown as a percentage referred to as the untreated control. The magnitude of the antibiofilm activity is categorized as follows: low (i.e., 10% < biofilm reduction < 25%), moderate (i.e., 25% ≤ biofilm reduction < 60%), high (i.e., 60% ≤ biofilm reduction < 90%), and excellent (i.e., biofilm reduction ≥ 90%).

## Data Availability

The data are contained within the article or the Appendix A.

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
