# Peer review of "Repurposing High-Throughput Screening Reveals Unconventional Drugs with Antimicrobial and Antibiofilm Potential Against Methicillin-Resistant Staphylococcus aureus from a Cystic Fibrosis Patient"

_antibiotics, 2025, doi:10.3390/antibiotics14040402_

Round 1
Reviewer 1 Report
Comments and Suggestions for Authors
This is a well-organized and important study that offers valuable insights into drug repurposing for MRSA infections in CF patients. However, to enhance the clarity, justification of findings, and discussion of limitations, a few revisions are needed. Addressing these points will strengthen the scientific accuracy and credibility of the manuscript
- The phrase "for the first time" is used several times, but previous studies have explored drug repurposing for MRSA. Could you clarify if this is the first time these specific drugs are being tested against MRSA? Or is it the first time their activity in the CF lung environment or antibiofilm properties are being explored? Please specify.
- In the "Identification of hits active against S. aureus biofilm formation" section, was the threshold of "≤ 10% growth inhibition" used to define non-antibacterial compounds determined experimentally, or was it based on prior literature? If it is from the literature, adding a reference would help support this claim.
- In Figure 3, could you specify what statistical test was used to determine that 193 hits caused a 100% inhibition rate? A brief clarification of the statistical method would be helpful.
- The readability of some figures (such as Figure 5 on biofilm dispersion) could be improved with clearer labels. This will help readers better interpret the data presented.
Reviewer 2 Report
Comments and Suggestions for Authors
The manuscript titled “Repurposing high-throughput screening reveals unconventional drugs with antimicrobial and antibiofilm potential against methicillin-resistant Staphylococcus aureus from a cystic fibrosis patient” presents very interesting and novel results which are of utmost importance considering the current antimicrobial resistance crisis.
Addressing the comments below should help increase the clarity of several points:
Please pay attention to spelling and typos eg.: edit “soya” to “soy” in Materials and Methods. Line 113, “Based” capitalization not needed.
In Materials and Methods, please indicate if biofilms were grown under aerobic or anaerobic conditions, shaking vs. static conditions.
To complement figure 1, can the authors specify the % development stages of different groups of drugs (x% of cancer drugs in Phase 1 etc..)?
Please elaborate if the concentrations used are physiologically relevant?
The antibiofilm compounds with excellent activity was initially 14 however after further testing, this number was reduced to 5 hits. Please indicate the difference in the biofilm set up techniques in the 2 approaches which led to the elimination of 9 candidates. Similarly, for the mature biofilm dispersion assay as well. Please clarify.
Can the authors discuss if the antibacterial (planktonic) activity observed is bactericidal or bacteriostatic? Did the authors plate aliquots from each well to investigate the potential growth of any remaining bacteria?
Reviewer 3 Report
Comments and Suggestions for Authors
- Please, describe which methods were used to identifying the compounds against CF patients.
- The manuscript is a bit confusing. I suggest that author focused more on the compounds which are used against S. aureus in CF patients.
- If S. aureus Sa2 strain is isolated by authors, it will be better describing identification part of the strain.
- If not, so please add information about the source of strain.
Round 2
Reviewer 2 Report
Comments and Suggestions for Authors
The authors addressed the majority of my comments.
Please use the materials and methods section to detail how the "cell viable count" measurements were made as this detail is missing.
Please correct the spelling mistakes such as soya vs soy as these are still seen in the edited version.
